# Lifestyle Factors Influencing Metabolic Syndrome after Adjusting for Socioeconomic Status and Female Reproductive Health Indicators: A National Representative Survey in Korean Pre- and Postmenopausal Women

**DOI:** 10.3390/healthcare12080821

**Published:** 2024-04-12

**Authors:** Eunyoung Hong, Youngmi Kang

**Affiliations:** 1Sustainable Health Research Institute, College of Nursing, Gyeongsang National University, Jinju 52828, Republic of Korea; dream@gnu.ac.kr; 2East-West Nursing Research Institute, College of Nursing Science, Kyung Hee University, Seoul 02447, Republic of Korea

**Keywords:** lifestyle, menopause, metabolic syndrome, reproductive health, socioeconomic status

## Abstract

Metabolic syndrome (MetS) is increasing markedly among postmenopausal women. Although studies suggest multiple risk factors for its development, few have investigated changes in socioeconomic status (SES), female reproductive health indicators (menarche age, experience of pregnancy, delivery, breastfeeding, and postmenopausal status), and lifestyle factors. This study investigated lifestyle factors affecting MetS prevalence among pre- and post-menopausal women after adjusting for SES and female reproductive health indicators. Data from the Korea National Health and Nutrition Examination Survey VII (2016–2018) on 2856 pre- and postmenopausal women aged 40–59 years were analyzed. Differences in SES (e.g., age, education, and household income), female reproductive health indicators (e.g., age of menarche and menopause), and lifestyle (e.g., total calorie intake, fats, and proteins, percentage of energy from carbohydrates, fats, and proteins, smoking, physical activity, and obesity) between MetS and non-MetS groups were calculated by performing χ^2^ or *t*-tests. Consequently, current smoking, physical inactivity, overweight, and obesity were significantly associated with increased MetS after adjusting for SES and female reproductive health indicators using logistic regression analysis. Hence, health policies and programs focusing on modifiable MetS risk factors–encouraging healthy eating habits, smoking cessation, and regular exercise—must be formulated to prevent the development of MetS in pre- and postmenopausal women.

## 1. Background

Metabolic syndrome (MetS) is a major public health challenge causing socioeconomic problems worldwide. It can be defined as a cluster of conditions characterized by obesity, dyslipidemia, hypertension, and glucose intolerance [1]. The prevalence of MetS in the United States increased dramatically from 37.6% during 2011–2012 to 41.8% during 2017–2018 [2]. The overall age-adjusted prevalence of MetS showed an increasing tendency from 27.1% in 2001 to 33.2% in 2020 in Korea [3].

The prevalence of MetS markedly increases in postmenopausal women [1]. Previous studies reported that multiple risk factors contributed to its development in this population [4]. However, few studies have investigated changes in potential contributors, such as socioeconomic status (SES), female reproductive health indicators, and lifestyle factors [5]. Specifically, previous studies found that MetS is positively associated with older age [1], city-dwelling [6], and high body mass index (BMI) [7,8]. Furthermore, MetS is positively associated with low education and income [9].

Given the detrimental consequences of MetS, understanding female reproductive health indicators associated with its development is important. Some studies found no relationship between MetS development and early menarche [10], high gravidity [11], and high parity [10,11], while others found a relationship between MetS development and early menarche [12], parity, and pregnancy [13]. A correlation was also found between early menarche and the likelihood of marriage at an early age, particularly in low-SES populations [14], which leads to pregnancy and childbirth and long-term risk factors for cardiovascular diseases [13]. Despite the long-term effects of female reproductive health indicators on MetS, few studies have investigated female reproductive health indicators associated with the disease. Menopausal state data included various types of menopausal states, but it did not exclude women who experienced early or artificial menopause [4].

Lifestyle factors—such as smoking, dietary intake, alcohol consumption, physical activity, and obesity—in patients with MetS have received considerable attention. Significant lifestyle changes have occurred in South Korea, including increased carbohydrate, fat, and alcohol consumption, physical inactivity, and excessive smoking [15]. These changes were major modifiable lifestyle risk factors for MetS in Koreans and resulted in an increased incidence of MetS, diabetes, cardiovascular disease, and cancer [16]. It has been reported that, as the percentage of carbohydrate energy intake increases, there tends to be a decrease in the concentration of HDL cholesterol in the bloodstream [17], potentially increasing the risk of MetS [18,19]. However, studies analyzing the role of lifestyle factors in postmenopausal women are scarcely found.

How SES mediates or modifies the relationship between lifestyle factors and the incidence of MetS in women remains unclear [16]. Additionally, few studies have investigated lifestyle factors in menopausal women and have obtained results after controlling for female reproductive health indicators [5]. Therefore, this study aimed to identify risk factors for the prevalence of MetS between MetS and non-MetS groups in pre- and postmenopausal women. It also investigated lifestyle factors influencing MetS prevalence after adjusting for SES and female reproductive health indicators in pre- and postmenopausal women with MetS via a nationally representative survey.

## 2. Methods

### 2.1. Design

This study performed a secondary analysis of data obtained from the Korea National Health and Nutrition Examination Survey (KNHANES) VII (2016–2018), conducted by the Ministry of Health and Welfare of the Republic of Korea.

### 2.2. Study Population

This population-based survey used multistage stratified cluster sampling and included three assessments: a health interview, a health examination, and a nutritional survey. Female participants aged 40–59 years were analyzed (n = 4056). Women who responded with “menstruating” to the query “Are you currently menstruating?” were categorized as premenopausal, while those who responded with “natural menopause” were classified as postmenopausal. Thus, we included 3416 participants who answered “menstruating (n = 1971)” or “natural menopause (n = 1445)”. Those whose responses indicated “pregnancy”, “postpartum/lactating”, or “artificial menopause” were excluded. Further, of the remaining participants, we excluded those who had premature menopause (age of menopause < 40 years, n = 17) or did not complete the assessment (n = 543). Finally, 2856 participants were included.

### 2.3. Definition of Metabolic Syndrome

MetS was defined by the American Association of Clinical Endocrinologists [20] and modified by the National Cholesterol Education Program, Adult Treatment Panel III [21], as the presence of three or more of the following: (1) abdominal obesity: waist circumference ≥ 85 cm based on the Korean Society for the Study of Obesity, (2) elevated serum triglycerides: ≥150 mg/dL, (3) reduced high-density lipoprotein (HDL) cholesterol: <50 mg/dL, (4) elevated blood pressure: mean systolic blood pressure ≥ 130 mmHg or diastolic blood pressure ≥ 85 mmHg or currently receiving treatment for hypertension, and (5) elevated fasting blood glucose levels: ≥100 mg/dL or currently using hypoglycemic agents or insulin. The health examination data were used to determine waist circumference, triglyceride, HDL cholesterol, blood pressure, and fasting blood glucose.

### 2.4. Measure

The measurements included SES, female reproductive health indicators, and lifestyle factors. SES included age, marital status, educational level, employment status, and household income. To distinguish between marital status, this variable was categorized into “married” and “unmarried”, with “unknown” or “no response” treated as missing values. Women who answered “housewife”, “student”, or “no” to the question, “Do you have a job?” were classified as “unemployed”. The others were classified as “employed”. Female reproductive health indicators included age at menarche, pregnancy experience, childbirth experience, breastfeeding experience, and menopausal status. The age of menarche was obtained by asking about the exact age in years, “At what age did you have your first period?” The pregnancy experience was determined only for individuals who responded with either “yes” or “no” to the question, “Do you have any experience with pregnancy?”, while “don’t know” and non-responses were treated as missing data. The childbirth experience was determined only for individuals who responded with either “yes” or “no” to the question, while “don’t know” and non-responses were treated as missing data. People who responded as currently menstruating were classified as premenopausal, while those who responded as experiencing natural menopause were categorized as postmenopausal.

Lifestyle factors included total calorie intake (kcal/day), total carbohydrates (g/day), total fats (g/day), total proteins (g/day), percentage of energy from carbohydrates, percentage of energy from fats, percentage of energy from proteins, smoking status, high-risk alcohol consumption, regular physical exercise, and obesity. The nutritional data were derived from the dietary intake survey conducted using the 24-h recall method in the nutritional survey section of the KNHANES. The dietary intake survey is conducted through a 24-h dietary recall via interview, prompting participants to recall their food consumption from the previous day. While this method has limitations in accuracy due to the reliance on recalling meals from the previous day, it holds significance as an excellent national statistic encompassing the entire population of South Korea [22]. Total calorie intake (kcal/day), total carbohydrates (g/day), total fats (g/day), and total proteins (g/day) were used as presented in the raw data without alteration. The percentage of energy derived from carbohydrates and proteins was computed by multiplying the total intake of carbohydrates and proteins (g/day) by four and then dividing it by the total caloric intake (kcal/day). Likewise, the percentage of energy from fats was determined by multiplying the total intake of fats (g/day) by nine and then dividing it by the total caloric intake (kcal/day).

Smoking status was classified as current smoker, ex-smoker, or non-smoker. High-risk drinking was classified as drinking more than five cups (three cups for beer) of soju or whisky more than twice a week. Each drink was calculated based on individual glasses, regardless of whether it was soju or whisky.

Regular physical exercise was classified using the Global Physical Activity Questionnaire, which assesses moderate to vigorous physical activity in three domains: work-related activities, activities during transportation, and leisurely activities. Total weekly minutes of moderate and vigorous activity in each domain were converted into metabolic equivalent task (MET) minutes per week. Individuals with a total of 600 MET minutes per week or more were classified as the regular physical exercise group.

Obesity was categorized according to BMI. Individuals with a BMI < 18.5 (kg/m^2^) were classified as underweight, those with a BMI ≥ 18.5 (kg/m^2^) but <23.0 (kg/m^2^) were classified as normal weight, those with a BMI ≥ 23.0 (kg/m^2^) but <25.0 (kg/m^2^) were categorized as overweight, and those with a BMI ≥ 25.0 were classified as obese [23].

### 2.5. Data Analysis

Data analysis was performed using SPSS software (version 27.0). Differences between the MetS and non-MetS groups regarding SES, female reproductive health indicators, and lifestyle factors were calculated by performing a chi-squared (χ^2^) test or *t*-test. The influence of lifestyle factors on MetS development was assessed using multiple logistic regression analysis, and the odds ratios and 95% confidence intervals were calculated after adjusting for SES and female reproductive health indicators.

### 2.6. Ethical Consideration

The Institutional Review Board of the Korea Centers for Disease Control and Prevention approved the protocols of this research and data release for the seventh KNHANES (2018-01-03-P-A). All participants provided written informed consent for collecting data before participating in the survey. This study was approved for exempt review by the Institutional Review Board of the first author’s affiliated university, Gyeongsang National University (IRB No.: *22-X-0027).

## 3. Results

MetS prevalence was 18.7% among 2856 participants. Differences in MetS composition between the MetS and non-MetS groups are shown in Table 1. Significant differences were observed in waist circumference, serum triglyceride levels, HDL cholesterol levels, systolic and diastolic blood pressure, and fasting blood glucose levels between the two groups.

Differences in SES, female reproductive health indicators, and lifestyle factors between the MetS and non-MetS groups are listed in Table 2. Regarding SES differences, 2322 (81.3%) did not have MetS (non-MetS group), while 534 (18.7%) had MetS. The mean age of the MetS group was 50.8 ± 5.8 years, which was significantly higher than that of the non-MetS group. Of the participants in the non-MetS group, 43.6% had higher than college education. Approximately 65.1% of the participants in the non-MetS group had a job, with no significant difference between the two groups. A significant difference was also observed in household income, at 32.2% in the fifth quintile for the non-MetS group and 21.6% in the fourth quintile for the MetS group.

In terms of female reproductive health indicators, the mean age of menarche in the MetS group was 13.8 ± 1.8 years, which was significantly higher than that observed in the non-MetS group. Of the participants, 98.3% experienced childbirth, while 75.4% experienced breastfeeding; no significant difference was observed between the two groups. In addition, 51.7% of the MetS group was menopausal, which was significantly higher than in the non-MetS group.

Regarding lifestyle factors, the total calorie intake in the MetS groups was 1635.8 ± 655.0 (kcal/day), which was significantly lower than in the non-MetS group. Furthermore, the total fat intake and the total protein intake were significantly lower in the MetS group than in the non-MetS group. The percentage of energy from carbohydrates in the MetS group was 66.1 ± 12.0%, which was significantly higher than in the non-MetS group. However, the percentage of energy from fats and the percentage of energy from proteins were significantly lower in the MetS group than in the non-MetS group; 7.7% of the MetS group were smokers, indicating a significant difference with the non-MetS group. In the MetS group, 6.9% were identified as high-risk drinkers, indicating a significantly higher prevalence compared to the non-MetS group. In addition, 36.3% of the MetS group was engaged in regular physical activity, indicating a significant difference with the non-MetS group.

Based on the aforementioned analysis, a logistic regression analysis was conducted, incorporating variables that exhibited significant differences between the non-MetS and MetS groups. Before conducting the analysis, we performed a linear regression analysis to calculate the variance inflation factor. Consequently, the variance inflation factor values ranged from 1.019 to 3.362, indicating no multicollinearity. The results of the logistic regression analysis with MetS as a dependent variable are shown in Table 3.

After adjusting for SES and female reproductive health indicators, lifestyle factors such as current smoking, physical inactivity, overweight, and obesity were identified as significantly influencing MetS. When compared to the non-smoking group, the current smoking group showed 2.465 times higher odds of MetS incidence. Similarly, the physically inactive group, in contrast to those participating in regular physical exercise, demonstrated 1.274 times higher odds of MetS occurrence, respectively. Compared to individuals with normal weight, those who were overweight had 3.550 times the odds of increased risk of MetS. Furthermore, individuals with obesity had a significantly higher risk, with 15.404 times the odds of increased risk of MetS. After adjusting for SES and female reproductive health indicators, additional stratification by menopausal status revealed that in premenopausal women, current smoking, overweight, and obesity were significant factors, whereas in postmenopausal women, physical inactivity, overweight, and obesity were identified as factors significantly influencing MetS (Appendix A).

## 4. Discussion

The KNHANES VII (2016–2018) was used to investigate the differences in SES, female reproductive health indicators, and lifestyle factors between the MetS and non-MetS groups in pre- and postmenopausal women. In addition, we investigated lifestyle factors that influenced MetS, adjusting for SES and female reproductive health indicators among pre- and postmenopausal women with MetS. Regarding the differences in SES between the MetS and non-MetS groups, the MetS group exhibited significantly higher mean age, lower education level, and lower household income than the non-MetS group. These results were consistent with those of previous studies [1,4,7,8,24]. Thus, it is important to strengthen MetS prevention education, with particular emphasis on younger women with lower SES; these women should be prioritized when implementing MetS prevention programs.

Regarding female reproductive health indicators, the menarche age of the MetS group was higher than that of the non-MetS group. Previous studies have demonstrated mixed results on the relationship between the development of MetS and the age of menarche. Studies have shown a relationship between an early age of menarche and a high risk of MetS [12,13,25]. According to a systematic review and meta-analysis, a 1-year decrease in age at menarche suggested an 8% increase in risk of MetS [25]. However, no correlation between age at menarche and MetS was found in other studies [1,11]. Therefore, additional studies are needed to examine the relationship between age of menarche and its potential role in decreasing or increasing MetS and its components among pre- and postmenopausal women.

Breastfeeding experience was not significantly associated with the development of MetS in the MetS group. This result is consistent with the findings of previous studies [26,27]. Ra and Kim [26] reported no significant association between a decreased likelihood of MetS and breastfeeding experience or duration in 1983 postmenopausal women, using data from the Korean Genome and Epidemiology Study. Conversely, Tørris and Bjørnnes [27] conducted a systematic review and meta-analysis on the association between breastfeeding and MetS, which suggested that lactation might play a protective role against MetS. In addition, Matsunaga et al. [28] reported that a lower probability of MetS was associated with the longest total duration of breastfeeding (>0 months) compared to those under 55 years with no history of breastfeeding. Further research is required to investigate lactation duration and its potential role in ameliorating or reversing MetS and its components. The present study observed a gradual increase in the incidence of MetS in the six years before and after the last menstrual period. Previous studies support the association between estrogen loss and central fat accumulation after menopause [3]. Therefore, initiating a prevention program in the premenopausal stage may be more effective in preventing MetS.

Regarding lifestyle factors, there was no difference in total carbohydrate intake between the two groups; however, the percentage of energy from carbohydrates was higher in the MetS group than in the non-MetS group, consistent with findings from previous studies [29,30]. Exceeding a certain energy percentage of carbohydrates increased the risk of MetS development [29]. According to the dietary reference intake of Koreans in 2020 [31], the recommended energy ratio of carbohydrates for women was set at 55–65% for all age groups. However, this was 66.41 ± 0.59% in the MetS group, exceeding the recommended amount. This result was consistent with those of previous studies [29,30]. Jung et al. [30] also reported that more than half of the participants aged 20 years in Korea had a carbohydrate energy intake of 70%. A comparative study with Americans and Koreans reported no significant association between the percentage of energy from carbohydrates and the incidence of MetS in Americans, whereas a significant association was observed in Koreans [30]. In Asian cultures, carbohydrates are a major source of dietary energy, potentially contributing to increased waist circumference and risk of MetS in Korean women [19].

We also found that total calorie intake, total fats, total proteins, the percentage of energy from fats, and the percentage of energy from proteins were lower in the MetS group than in the non-MetS group. Previous studies showed mixed results concerning the association of daily calorie intake and fat and protein intake with the development of MetS in women. Lee et al. [32] found that daily calorie, fat, and protein intake were significantly higher in middle-aged Korean women with MetS. Vasbinder et al. [33] also reported that higher total protein intake, specifically animal protein, was strongly associated with MetS in postmenopausal women. Conversely, Jamshidi et al. [34] reported that a higher intake of total protein is associated with lower odds of having MetS in Iranian women. Lee et al. [35] investigated the correlations between high fat and high protein with MetS using the KNHANES (2018–2020), but no relationship was found between them. Further research is needed to investigate the effects of specific dietary patterns on the development of MetS in pre- and post-menopausal women. As menopause is inevitable, a balanced diet with adequate nutrition—which includes carbohydrates, proteins, polyunsaturated fatty acids, riboflavin, and calcium—is recommended for women likely to obtain energy from carbohydrates. Therefore, promoting the awareness and use of nutritional labeling when developing strategies for target populations is important.

The association between smoking and MetS in females remains controversial. Smoking rates were significantly higher in the MetS group than in the non-MetS group, which was consistent with previous findings [8,36]. Liang et al. [8] reported a positive relationship between smoking and MetS in postmenopausal women in Southern China. Smoking cessation was the most effective way to reduce the risk of MetS and cardiovascular diseases [37]. However, smoking cessation was also associated with a high risk of MetS owing to subsequent weight gain [38].

Consistent with this, Kim [15] analyzed data from the Korean Medical Panel from 2013 to 2016 and reported that an increase in cigarette prices led to a decrease in smoking; however, smoking cessation led to an increase in weight by 3.09 kg and BMI by 1.3. Onat et al. [37] also reported that excessive smoking prevented future MetS in Turkish women by preventing an increase in waist circumference. Preventing weight gain after quitting smoking is also important to prevent MetS. From 2001 to 2015, smoking rates fluctuated in women, whereas female obesity rates decreased and subsequently increased again in Korean adults [38]. However, few studies have investigated the relationship between smoking and obesity in women. Therefore, further studies are required to clarify the relationships between smoking, BMI, and MetS in women.

A higher rate of regular physical activity was observed in the non-MetS group than in the MetS group. This result was supported by those of previous studies [8,39]. A sedentary lifestyle was a risk factor for MetS development in postmenopausal women. A systematic review and meta-analysis found that regular physical activity effectively reduced the risk of MetS in postmenopausal women, with significant metabolic improvement over 8–10 weeks of intervention [39]. Hence, regular physical activity was a practical strategy to reduce the occurrence of MetS. Exercise is a cost-effective intervention for preventing and ameliorating the effects of MetS; however, it remains underutilized. Therefore, including interventions that promote regular physical activity is important when developing future MetS prevention programs for pre- and postmenopausal women.

The regression analysis in the current study showed that lifestyle factors, such as current smoking, lack of physical activity, and obesity, increased the likelihood of MetS, after controlling for SES (age, education, and household income) and female reproductive health indicators (age of menarche and menopause). Smoking significantly doubled the likelihood of developing MetS. Lack of evidence for covariate effects of SES, female reproductive health indicators, or lifestyle factors on the likelihood of developing MetS limits the interpretations; however, these findings were inconsistent with those of a previous study [36]. Onat et al. [37] reported a positive relationship between heavy smoking and a reduced risk of MetS, after controlling for SES (age and basic family income) and lifestyle factors (physical activity rating) in Turkish women. The current study’s findings might be because the sample had a lower percentage of specific factors (e.g., unmarried and never been pregnant). Smoking played a protective role against the development of MetS and diabetes mellitus in Turkish women, primarily by protecting against obesity. To the best of our knowledge, this study was the first in Korea to report an association between smoking and MetS risk in pre- and postmenopausal women, after adjusting for SES and female reproductive health indicators. Further analyses may present additional meaningful implications.

### 4.1. Limitations

This study is significant because it was based on high-quality, nationally representative data that included and focused on women with various risk factors for MetS. Despite this advantage, the study had several limitations. First, we used only one 24-h recall, which may not represent the typical intake for dietary estimation. Thus, at least two 24-h recalls are needed for more reliable results. Second, recall bias may have existed in terms of dietary intake and physical activity. Third, confounding variables, such as other dietary characteristics, exercise frequency, and number of births, may not have been considered. Fourth, the KNHANES utilized a complex, stratified, multistage, probability-cluster survey design, which might have resulted in homogenous and disproportionate sampling (e.g., oversampling or adjustment for non-response). This may have resulted in underestimation and an increased probability of a type I error [40].

### 4.2. Implications

Our findings have implications for public health and policy. Healthy lifestyle changes to prevent and control MetS should be a major goal of national public health interventions. Currently, domestic healthcare policies focus on smoking cessation rather than obesity. Therefore, when anti-smoking policies are strengthened, an active healthcare policy is required to lower the obesity rate. Considering that smoking, physical inactivity, and obesity are the main preventable causes of MetS, our findings may provide an impetus for the development of strategies and the implementation of effective clinical and public health interventions to improve cardiometabolic health.

## 5. Conclusions

This study demonstrated that SES (higher mean age, lower education level, and lower household income) and female reproductive health indicators (age of menarche and menopause) were associated with the development of MetS in Korean women. In addition, we found that current smoking, physical inactivity, and obesity played important roles in MetS development after adjusting for SES and female reproductive health indicators in pre- and postmenopausal Korean women. We propose the development and implementation of health policies and public health programs focusing on modifiable risk factors for MetS, such as smoking cessation, regular exercise, and healthy eating habits to prevent and treat MetS.

## Figures and Tables

**Table 1 healthcare-12-00821-t001:** Metabolic syndrome components.

Components	Total	Non-MetS Group	MetS Group	t (*p*)
	N= 2856	n = 2322	n = 534	
	M ± SD	M ± SD	M ± SD	
Waist circumference (≥85 cm)	77.9 ± 9.0	75.7 ± 7.5	87.5 ± 8.7	−29.010 (0.000)
Triglyceride (≥150 mg/dL)	114.4 ± 82.4	95.6 ± 52.9	196.6 ± 126.1	−18.154 (0.000)
HDL cholesterol (<50 mg/dL)	55.7 ± 13.0	58.2 ± 12.4	45.0 ± 9.6	26.903 (0.000)
SBP (≥130 mmHg or medication)	114.4 ± 15.5	111.9 ± 14.3	125.2 ± 16.1	−17.557 (0.000)
DBP (≥85 mmHg or medication)	75.5 ± 9.4	74.1 ± 8.8	81.3 ± 9.6	−15.894 (0.000)
FBG (≥100 mg/dL or medication)	97.7 ± 23.0	93.5 ± 13.3	115.7 ± 40.7	−12.443 (0.000)
Number of components of the MetS	1.3 ± 1.3	0.8 ± 0.8	3.5 ± 0.7	−83.741 (0.000)

Note: HDL, High-density lipoprotein; SBP, Systolic blood pressure; DBP, Diastolic blood pressure; FBG, Fasting blood glucose; M, Mean; MetS, Metabolic syndrome; SD, Standard deviation.

**Table 2 healthcare-12-00821-t002:** Participants’ socioeconomic status, female reproductive health indicators, and lifestyle factors.

Variables	Categories	Total	Non-MetS Group	MetS Group	χ^2^ or t (*p*)
		n = 2856	n = 2322	n = 534	
		n (%) or M ± SD	n (%) or M ± SD	n (%) or M ± SD	
Socioeconomic status					
Age		49.3 ± 5.9	49.0 ± 5.9	50.8 ± 5.8	−6.450 (0.000)
Marital status	Yes	2761 (96.7)	2240 (96.7)	521 (97.7)	1.625 (0.202)
	No	95 (3.3)	82 (3.5)	13 (2.3)	
Education	Middle school or less	480 (16.8)	326 (14.0)	154 (28.9)	88.904 (0.000)
	High school	1222 (42.8)	984 (42.4)	238 (44.7)	
	College or higher	1153 (40.4)	1012 (43.6)	141 (26.5)	
Employment status	Employed	1836 (64.3)	1510 (65.1)	326 (61.0)	3.084 (0.079)
	Unemployed	1018 (35.7)	810 (34.9)	208 (39.0)	
Household income	First quintile	195 (6.8)	139 (6.0)	56 (10.5)	43.622 (0.000)
	Second quintile	453 (15.9)	339 (14.6)	114 (21.4)	
	Third quintile	587 (20.6)	469 (20.2)	118 (22.1)	
	Fourth quintile	758 (26.5)	628 (27.0)	130 (24.4)	
	Fifth quintile	862 (30.2)	747 (32.2)	115 (21.6)	
Female reproductive health indicators					
Age of menarche		13.7 ± 1.6	13.7 ± 1.6	13.8 ± 1.8	−2.334 (0.020)
Pregnancy	Yes	2748 (96.2)	2230 (96.0)	518 (97.0)	1.113 (0.291)
	No	108 (3.8)	92 (4.0)	16 (3.0)	
Childbirth experience	Yes	2700 (98.3)	2190 (98.2)	510 (98.5)	1.191 (0.275)
	No	48 (1.7)	40 (1.8)	8 (1.5)	
Breastfeeding experience	Yes	2153 (75.4)	1739 (74.9)	414 (77.5)	1.625 (0.202)
	No	703 (24.6)	583 (25.1)	120 (22.5)	
Menopause	Yes	1159 (40.6)	883 (38.0)	276 (51.7)	33.586 (0.000)
	No	1697 (59.4)	1439 (62.0)	258 (48.3)	
Lifestyle factors					
Total calories intake (kcal/day)		1717.4 ± 676.2	1736.1 ± 697.7	1635.8 ± 655.0	3.088 (0.002)
Total carbohydrates (g/day)		272.4 ± 114.3	273.8 ± 114.9	266.3 ± 111.2	1.369 (0.171)
Total fats (g/day)		38.3 ± 25.6	39.3 ± 25.9	33.8 ± 23.5	4.454 (0.000)
Total proteins (g/day)		61.8 ± 30.1	62.8 ± 29.9	57.2 ± 30.4	3.912 (0.000)
Percentage energy of carbohydrates		64.2 ± 12.0	63.8 ± 11.9	66.1 ± 12.0	−4.086 (0.000)
Percentage energy of fats		19.6 ± 8.6	19.9 ± 8.6	18.1 ± 8.6	4.510 (0.000)
Percentage energy of proteins		14.4 ± 4.1	14.5 ± 4.1	14.0 ± 4.0	2.913 (0.004)
Smoking	Current smoker	132(4.6)	91(3.9)	41 (7.7)	14.120 (0.001)
	Ex-smoker	145 (5.1)	118 (5.1)	27 (5.1)	
	Non-smoker	2572 (90.3)	2109 (91.0)	463 (87.2)	
High-risk drinking	Yes	149 (5.2)	112 (4.8)	37 (6.9)	3.896 (0.048)
	No	2702 (94.8)	2206 (95.2)	496 (93.1)	
Regular physical exercise	Yes	1257 (44.0)	1063 (45.8)	194 (36.3)	15.797 (0.000)
	No	1598 (56.0)	1258 (54.2)	340 (63.7)	
Obesity	Underweight	95 (3.4)	93 (4.1)	2 (0.4)	645.326 (0.000)
	Normal	1354 (48.8)	1284 (56.8)	70 (13.7)	
	Overweight	564 (20.3)	473 (20.9)	91 (17.8)	
	Obese	760 (48.8)	412 (18.2)	348 (45.8)	

Note: M, Mean; SD, Standard deviation; Non-MetS, Non-metabolic syndrome; MetS, Metabolic syndrome.

**Table 3 healthcare-12-00821-t003:** Adjusted odds ratio of socioeconomic status, female reproductive health indicators, lifestyle factors, and metabolic syndrome.

Variables	Categories	OR (95% CI)	*p*
Socioeconomic status			
Age	-	1.000 (0.966–1.035)	0.999
Education	Middle school or less	1.343 (0.925–1.949)	0.121
	High school	1.118 (0.849–1.471)	0.428
	College or higher	1	
Household income	First quintile	1.656 (1.056–2.598)	0.028
	Second quintile	1.380 (0.973–1.956)	0.071
	Third quintile	1.064 (0.800–1.575)	0.502
	Fourth quintile	1.064 (0.772–1.465)	0.706
	Fifth quintile	1	
Female reproductive health indicators			
Age of menarche	-	1.027 (0.956–1.104)	0.466
Menopause	Yes	1.425 (0.978–2.075)	0.065
	No	1	
Lifestyle factors			
Total calories (kcal/day)	-	1.000 (0.999–1.000)	0.505
Total fats (g/day)	-	0.995 (0.979–1.012)	0.587
Total proteins (g/day)	-	1.004 (0.999–1.000)	0.556
Percentage of energy from carbohydrates	-	1.005 (0.986–1.025)	0.593
Percentage of energy from fats	-	1.000 (0.961–1.041)	0.999
Percentage of energy from proteins	-	0.968 (0.901–1.039)	0.366
Smoking	Current smoker	2.465 (1.496–4.063)	0.000
	Ex-smoker	1.136 (0.688–1.875)	0.619
	Non-smoker	1	
High-risk drinking	Yes	0.981 (0.582–1.653)	0.943
	No	1	
Regular physical exercise	Yes	1	
	No	1.274 (1.014–1.601)	0.038
Obesity	Underweight	0.367 (0.087–1.539)	0.170
	Normal	1	
	Overweight	3.550 (2.515–5.011)	0.000
	Obese	15.404 (11.430–20.759)	0.000

Note: CI, Confidence interval; OR, Odds ratio.

## Data Availability

The datasets used and/or analyzed during the current study are available from the corresponding author upon reasonable request.

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
