# Peer review of "Lifestyle Factors Influencing Metabolic Syndrome after Adjusting for Socioeconomic Status and Female Reproductive Health Indicators: A National Representative Survey in Korean Pre- and Postmenopausal Women"

_healthcare, 2024, doi:10.3390/healthcare12080821_

Round 1

Reviewer 1 Report (Previous Reviewer 2)

Comments and Suggestions for Authors

I have no further comments.

Best wishes

Author Response

Supplemental materials were provided. Thanks.

Reviewer 2 Report (Previous Reviewer 1)

Comments and Suggestions for Authors

Compared to the previous submission, there are substantial changes. 

However, some small issues remain:

1. Lines 154-157 "Lifestyle factors that affected MetS were assessed by performing complex sample logistic regression analysis with weighting.

Please, perform logistic regressions without weighting.

As I said before, the weighting is not good for logistic regressions, because it introduces errors in estimating CI (and p-value), because it uses the whole population size instead of the actual sample size to estimate the statistical significance.

Therefore, you will have the inflation of significance, since the whole sample size will be 1000 higher. (Unweighted (N=2,856); Weighted (N=6,787,918)!)

"Using the unweighted data can lead to biased estimates, especially when the sampling weights vary significantly from one case to the other (note: your weighted and non-weighted data are very similar, according to Tables 1 and 2, so sampling weights are not necessary). On the other hand, although applying the simple frequency weights can predict unbiased estimates, it would underestimate the standard errors, which in turn would cause every test to be significant, because the sample size is considered as large as the population."

https://www.ncbi.nlm.nih.gov/pmc/articles/PMC3217522/

2. Additionally, Tables 1 and 2: please, eliminate weighted results. It seems that you have total N=6,787,918 (the whole population), but you have only a sample of 2,856.

Only data based on the actual sample site should be presented and analyzed.

(Comment: weighting is useful if the goal of your research is to describe the population, and see the prevalence of something in the whole population. For example, you would like to see the prevalence of smoking in the whole population, so you can weight cases according to their sex and age, to achieve a "more representative" sample than the one you have actually in your research. 

In regression analyses, it is used only when you have a violation of homoscedasticity (i.e., when you have heteroscedasticity), or when you have an imbalanced data set (e.g., when N-Mets is more than 100 times lower number than N-NonMets). In your case, your Mets group is just 4-5 times smaller than NonMets, so you do not have an imbalanced data set. This weighing is a completely different statistical approach from what you have used for your research).

https://towardsdatascience.com/weighted-logistic-regression-for-imbalanced-dataset-9a5cd88e68b

https://www.sciencedirect.com/science/article/abs/pii/S0950705114000239)

3. As I said, the menopausal status stratification needs to be performed. In Table 2 you have a significant difference in menopausal status between Mets and Non-Mets. Even though you did not have a statistical significance of menopause in regression analyses for OR for Mets (Table 3), it would be interesting to see if the same factors are equally important for Mets development in post-menopausal and pre-menopausal women (and your paper is focussed on reproductive health). Are smoking, obesity, and physical activity equally significant in both post-menopausal and pre-menopausal women groups?

So, simply divide your data set into 2 data sets, and perform unweighted regression analyses separately for both of them, to see if you will have any difference in your findings. You do not have to put those results as new Tables. You can send them here, as attachments, to see, or give them as supplementary material, but you can describe them in the manuscript text, depending on the findings  (e.g., "Additional stratification by menopausal status did not change our findings (data not shown)", or e.g., "Additional stratification by menopausal status showed that in postmenopausal women the most significant factors were....  while in premenopausal women the most significant factors were....")

This will add additional value to your manuscript and make the discussion more interesting.

(Of course, since the data sets will be smaller for the stratified analyses, maybe some p-values will lose now significance. But, let's first see what you will obtain in the stratification analyses.)

4. Not only obesity but also overweight is an important factor for Mets (according to Table 3), so in the abstract and manuscript this should be corrected

E.g., lines 22-25: "Consequently, current smoking, physical inactivity, and overweight /obesity were significantly associated with increased MetS after adjusting for SES and female reproductive health."

Specific changes in the text:

5. In the abstract, line 22, please include regression analyses: "by performing χ2 or t-tests and logistic regression".

6. lines 13, 19, 24, 213, 304... (and everywhere in the text where applicable): It should be: "female reproductive health indicators"

7. Lines 15-16: It should be:  This study investigated lifestyle factors affecting MetS prevalence among pre- and post-menopausal women by adjusting for SES and female reproductive health indicators.

8. Line 16: Please, eliminate: "Using multistage stratified sampling on". It should be "Data from the Korea National Health and Nutrition Examination Survey VII (2016–2018) on 2,856 pre- and postmenopausal women aged 40–59 years were analyzed..."

9. Line 22: instead of "current smoker" it should be "current smoking" everywhere in the body text (except line 135).

10. Line 26: instead of "deal with" use "prevent development of"

11. Line 37: "has markedly increased" change with "markedly increases"

12. 37-39: Previous studies reported that multiple risk factors contributed to its development in this population." - please give a reference that supports this statement.

13. 42-43: Positively, not negatively (see reference 8). "Furthermore, MetS was positively associated with low education and income [8];"

14: line 44: references 9 and 10 are not referred to this sentence.

15. Reference 24. Give instead the link in English: https://www.jomes.org/journal/view.html?doi=10.7570/jomes21022

16. lines 154-157: Please, change to "The influence of lifestyle factors on MetS development were assessed by multiple logistic regression analysis, and the odds ratios and 95% confidence intervals were calculated after adjusting for the above-mentioned SES and female reproductive health indicators."

17. lines 167-171: Please eliminate t and p values (there should not be a repetition of data from Table). It should be:

"Significant differences were observed in waist circumference, serum triglyceride level, HDL cholesterol level, systolic and diastolic blood pressure, and fasting blood glucose level between the two groups."

18. The same: lines 179-186: Please eliminate t, X2 and p values (there should not be a repetition of data from Table). 

19. Results text and Tables 1 and 2: Please, give 1 decimal place for mean and SD (the same as for %)

20. Table 2 notes: Was it instead meant "Significant also after the Bonferroni correction"? Please, clarify.

21. The same: lines 191-206: Please eliminate mean+-SD, t, X2 and p values (there should not be a repetition of the numeric data from Table). 

22. Line 207: please, eliminate "complex sample"

23. The discussion needs to be rewritten because it refers to the previous version of this manuscript:

Line 229: "Higher carbohydrate energy ratio, current smoking, and physical inactivity were significantly associated with increased MetS, after adjusting for SES..." - this is not found in this manuscript.

254-267: However, now you did not find the significant influence of % carbohydrates on Mets in regression analyses.

24. Line 268-269: "We also found that total fats, total proteins, the percentage energy of fats, and the percentage energy of proteins were lower in the MetS group than in the non-MetS group." - please, give some references or other research on that and compare your results with theirs.

25. Line 275-281: Please, make stratified analyses as required above.

26. Line 302: "current smoking"

27. Line 323. Instead "for future research" use "for more reliable results"

28. Lines 323-324. Please, change it to "such as other dietary characteristics"

29. "Third, recall bias may have existed in dietary intake and physical activity." Please, make this a second, after talking about 24 recall. (i.e., exchange the second and the third limitation).

30. Lines 329-330: Please, eliminate: "Fifth, since the analysis did not account for all macronutrients, caution is needed in interpreting the results." - now you analyzed all macronutrients

31: Lines 336-337 Please correct "Considering that a high energy ratio from carbohydrates," according to the novel findings

32: Lines 344-345: Please, eliminate  "a high percentage energy of carbohydrates, ", according to the novel findings.

Lines 343 and 347: female reproductive health indicators

Comments on the Quality of English Language

English language refining is needed.

Author Response

Supplemental materials were provided. Thanks.

This manuscript is a resubmission of an earlier submission. The following is a list of the peer review reports and author responses from that submission.

Round 1

Reviewer 1 Report

Comments and Suggestions for Authors

The authors have not made significant improvements, only small more typo-style corrections, and the main issues remain.

As suggested previously, in my previous report (manuscript ID healthcare-2770487) protein and fats must be included in the analyses (both in absolute terms and as % of total energy intakes).  Fats and proteins in the diet are also important predictors for METs development, and there is no acceptable explanation to be excluded from the analysis, particularly since there are data on fat and protein intake in the analyzed surveys. The authors again refused to add those additional variables to the analyses.

https://knhanes.kdca.go.kr/knhanes/eng/index.do

Please, see some references for macronutrient diet composition and METs:

Park H, Kityo A, Kim Y, Lee SA. Macronutrient Intake in Adults Diagnosed with Metabolic Syndrome: Using the Health Examinee (HEXA) Cohort. Nutrients. 2021 Dec 14;13(12):4457. doi: 10.3390/nu13124457. PMID: 34960009; PMCID: PMC8706324.

Hoyas I, Leon-Sanz M. Nutritional Challenges in Metabolic Syndrome. J Clin Med. 2019 Aug 24;8(9):1301. doi: 10.3390/jcm8091301. PMID: 31450565; PMCID: PMC6780536.

Kim JH, Lim JS. Prevalence Trends of Metabolic Syndrome among Korean Children and Adolescents from a Population-Based Cross-Sectional Survey. Life (Basel). 2022 Sep 9;12(9):1404. doi: 10.3390/life12091404. PMID: 36143440; PMCID: PMC9503497.

Kweon S, Kim Y, Jang MJ, Kim Y, Kim K, Choi S, Chun C, Khang YH, Oh K. Data resource profile: the Korea National Health and Nutrition Examination Survey (KNHANES). Int J Epidemiol. 2014 Feb;43(1):69-77. doi: 10.1093/ije/dyt228. PMID: 24585853; PMCID: PMC3937975.

 Additionally, BMI must be placed in the lifestyle factors (not socio-economic factors!), since the authors analyze here the lifestyle factors (particularly related to diet and physical activity, and BMI is strongly influenced by diet and physical activity). (please, see my previous report) So, there is no acceptable explanation "The other authors made also this". In some studies, that are not focused on lifestyle factors, is probably acceptable to put BMI in socioeconomic variables, but since we have here category "lifestyle factors", it is more correct to place BMI there, not in socioeconomic factors.  Even in the Korea National Health and Nutrition Examination Survey anthropometry is not classified as a "socioeconomic variable".

Kweon S, Kim Y, Jang MJ, Kim Y, Kim K, Choi S, Chun C, Khang YH, Oh K. Data resource profile: the Korea National Health and Nutrition Examination Survey (KNHANES). Int J Epidemiol. 2014 Feb;43(1):69-77. doi: 10.1093/ije/dyt228. PMID: 24585853; PMCID: PMC3937975.

Paek KW, Chun KH. Moderating effects of interactions between dietary intake and socioeconomic status on the prevalence of metabolic syndrome. Ann Epidemiol. 2011 Dec;21(12):877-83. doi: 10.1016/j.annepidem.2011.07.006. Epub 2011 Sep 1. PMID: 21889360.

Again, only one 24-hour recall is not adequate for dietary estimations. since it does not have to represent the usual intake, which has to be mentioned in the study limitations (at least two 24-hour recalls are needed).

Moreover, stratified analyses according to menopausal status should be performed since X2 results are significant (p= 0.000, Table 2), and again the authors refused to make those additional analyses. Menopause is a significant predictor for the development of METs in women, there is enough literature confirming that already. Hormonal changes strongly influence the development of METs after menopause. 

Lobo RA. Metabolic syndrome after menopause and the role of hormones. Maturitas. 2008 May 20;60(1):10-8. doi: 10.1016/j.maturitas.2008.02.008. Epub 2008 Apr 14. PMID: 18407440.

Again, the authors did not give all OR for covariates, as previously already suggested  (even they were asked to give them at least as supplementary material). It is not clear how variables were chosen for adjustment factors 

Correction in p-value for multiple testing is recommended in Table 2, which has not been done.

Results (text) are a repetition of the numbers given in the tables.

Seems that the discussion is again only a comparison with other, already well-known results, without any mechanisms given. Only small changes were introduced.

Comments on the Quality of English Language

No comments.

Reviewer 2 Report

Comments and Suggestions for Authors

Thank you for asking me to review Lifestyle Factors Influencing Metabolic Syndrome After Adjusting for Socioeconomic Status and Female Reproductive Health: A National Representative Survey in Korean Pre- and Postmenopausal Women again. 

It was a real pleasure to read this paper for the 3rd time and you are to be congratulated on persevering and drafting a very important piece of work.

There are just a couple of minor edits that you should address:

Page 3 line 114: The pregnancy experience was determined by yes,” “no,” “unknown,or noresponses. The childbirth experience was determined by responses of yes,” “no,” “not applicable,” “unknown,or noresponse.

Reviewer:

There are 2 ‘no’ responses. I don’t quite understand this. I suspect that the second ‘no’ means that this field wasn’t completed. If so, you need to make this clearer. 

Page 8 line 287 - (e.g., unmarried and neve been pregnant)

Reviewer: The R is missing in ‘never’  

Page 8 Line 319: and female reproductive health (longer breastfeeding duration and menopause) were associated with the development of MetS in pre- and postmenopausal Korean women.

Reviewer: This is confusing, how can menopause be associated with higher MetS in pre-menopausal women.

Well done, team!